# A Vehicle Trajectory Privacy Preservation Method Based on Caching and Dummy Locations in the Internet of Vehicles

**DOI:** 10.3390/s22124423

**Published:** 2022-06-11

**Authors:** Qianyong Huang, Xianyun Xu, Huifang Chen, Lei Xie

**Affiliations:** 1College of Information Science and Electronic Engineering, Zhejiang University, Hangzhou 310027, China; 22031074@zju.edu.cn (Q.H.); 21831091@zju.edu.cn (X.X.); xiel@zju.edu.cn (L.X.); 2Zhoushan Ocean Research Center, Zhoushan 316021, China; 3State Key Laboratory of Fluid Power and Mechatronic Systems, Zhejiang University, Hangzhou 310027, China; 4Zhejiang Provincial Key Laboratory of Information Processing, Communication, and Networking, Hangzhou 310027, China

**Keywords:** internet of vehicles (IoVs), continuous location-based service (LBS), trajectory privacy preservation, caching, dummy location

## Abstract

In the internet of vehicles (IoVs), vehicle users should provide location information continuously when they want to acquire continuous location-based services (LBS), which may disclose the vehicle trajectory privacy. To solve the vehicle trajectory privacy leakage problem in the continuous LBS, we propose a vehicle trajectory privacy preservation method based on caching and dummy locations, abbreviated as TPPCD, in IoVs. In the proposed method, when a vehicle user wants to acquire a continuous LBS, the dummy locations-based location privacy preservation method under road constraint is used. Moreover, the cache is deployed at the roadside unit (RSU) to reduce the information interaction between vehicle users covered by the RSU and the LBS server. Two cache update mechanisms, the active cache update mechanism based on data popularity and the passive cache update mechanism based on dummy locations, are designed to protect location privacy and improve the cache hit rate. The performance analysis and simulation results show that the proposed vehicle trajectory privacy preservation method can resist the long-term statistical attack (LSA) and location correlation attack (LCA) from inferring the vehicle trajectory at the LBS server and protect vehicle trajectory privacy effectively. In addition, the proposed cache update mechanisms achieve a high cache hit rate.

## 1. Introduction

The vehicular ad-hoc network (VANET) has become an important part of future intelligent transport systems. It will be widely applied in traffic management [1], road safety [2], information dissemination to drivers [3], etc. As more and more vehicles connect to the internet of things, the conventional VANET is developing into the internet of vehicles (IoVs).

Moreover, many mobile devices and applications (apps) use location-based services (LBS) applications [4]. As a user acquires the LBS, the location should be provided so that the location’s privacy is disclosed. In the IoVs, a vehicle user may act as the provider of location services. When participating in a task of swarm intelligence, a vehicle user will expose the location privacy. Hence, location privacy preservation in the LBS is an important problem to be solved [5].

In IoVs, vehicle users only need to provide the location information once to obtain a single LBS, such as “to inquire about a gas station nearby”. As vehicle users want to acquire continuous LBS, such as “to inquire real-time traffic feedback based on the current location”, they should provide the location information continuously.

However, many location privacy preservation methods only protect the privacy of a single location, and the privacy of the user’s trajectory may be disclosed if they use the continuous LBS. When the vehicle user utilizes LBS consecutively, and the spatial anonymity technology is adopted for the location privacy preservation, the user’s locations will be replaced by a series of anonymous areas. However, combined with the background knowledge or related technologies, the attacker can obtain the user’s trajectory information with high probability. Therefore, it is of great significance to protect the user’s trajectory privacy in the continuous LBS.

To protect the trajectory privacy of vehicle users as well as guarantee the quality of LBS, some trajectory privacy preservation methods, namely the dummy locations-based trajectory privacy preservation methods [6,7,8,9,10,11,12,13], trajectory anonymity-based privacy preservation methods [14,15,16], obfuscation-based privacy preservation methods [17,18,19,20,21,22], caching-based trajectory privacy preservation methods [23,24,25,26,27,28,29], and mixed zone-based privacy preservation method [30,31,32], have been proposed. However, there are still some problems with these methods. First, for the trajectory privacy preservation methods based on dummy locations [6,7,8,9,10,11,12,13], it is difficult to resist the long-term statistical attack (LSA) and/or the location correlation attack (LCA) in the continuous LBS. The problem of trajectory privacy preservation is prominent. Second, some trajectory privacy preservation methods based on caching mechanisms [23,24,25,26,27,28] rely on users’ collaborative caching without a reliable third party. However, due to the high mobility of vehicles, the validity of the cache cannot be guaranteed. If a vehicle user can obtain complete information by collaborating with other vehicles, the communication overhead and the risk of privacy exposure increase. Hence, an active cache deployment scheme was proposed to achieve a high cache hit rate [29]. However, in the proposed scheme, the complexity of the cache mechanism is high, and the issue of trajectory privacy preservation is not taken into consideration. Therefore, to deal with these issues, we investigate the problem of vehicle trajectory privacy preservation in IoVs in this work.

In this paper, we propose a vehicle trajectory privacy preservation method based on caching and dummy locations. The basic idea of our proposed method is to utilize the roadside units (RSUs) in the architecture of the IoV system to cache the hotspot data pushed and requested data sent by the LBS server. When a vehicle user requests the continuous LBS, combining the privacy preservation method based on dummy locations with the cache mechanism at the RSUs is used to protect the vehicle trajectory privacy. In addition, we address the cache update mechanisms based on dummy locations and timeliness to guarantee a high cache hit rate.

The main contributions of this paper are as follows.

The problem of vehicle trajectory privacy preservation in IoVs is studied. And a vehicle trajectory privacy preservation method based on caching and dummy locations is proposed. When a vehicle user requests the continuous LBS, the location privacy preservation method based on dummy locations is used to protect the vehicle location. And the caching deployed at the RSU with active and passive cache update mechanisms is utilized to protect the vehicle trajectory. Compared with the exiting dummy location-based methods [9,10], our proposed method deploys the caching at RSU so that both the RSU and LBS server cannot obtain all the data since a single RSU can only cover a part of the area. Moreover, the proposed method still applies dummy locations to protect trajectory privacy within a single RSU region. Hence, our proposed method can preserve trajectory privacy under LSA and LCA.Compared with the caching-based trajectory privacy preservation methods relying on users’ collaborative caching [27,28], this paper addresses the cache update mechanisms based on data popularity and dummy locations to protect location privacy, as well as improving the cache hit rate. In the initialized and periodic cache update phases, RSUs cache the hotspot data pushed by the LBS server. In the service-providing phase, RSUs cache the requested data sent by the LBS server. However, for the collaborative caching mechanism in [27,28], the validity of the cache cannot be guaranteed due to the high mobility of vehicles; the communication among users increases the risk of privacy exposure.The performance of the proposed vehicle trajectory privacy preservation method, in terms of security, computation overhead, communication overhead, and storage overhead at the RSUs, is analyzed. Furthermore, extensive simulations are conducted to evaluate the performance of the proposed method. The results show that the proposed method achieves a lower trajectory privacy disclosure probability and a higher cache hit rate.

The rest of the paper is organized as follows. The related work on location privacy preservation methods is overviewed in Section 2. In Section 3, some preliminaries and the problem to be solved in this paper are described. In Section 4, a vehicle trajectory privacy preservation method based on caching and dummy locations is presented in detail. Performance analysis and simulation results are given in Section 5. Finally, we conclude the paper in Section 6.

## 2. Related Work

The problem of location privacy preservation has been attracting wide attention from both academia and industry, and this problem draws even more attention due to the booming of LBSs. Up to now, many location privacy preservation methods have been proposed, including K-anonymity-based method [33,34,35,36], obfuscation-based method [37,38,39], differential privacy-based method [40,41], homomorphic encryption-based method [42,43,44] and dummy location-based method [6,7,8,9,10,11,12,13,45]. In this work, we focus on the trajectory privacy preservation method based on dummy locations in IoVs.

The trajectory privacy preservation for the continuous LBS is a hot research topic in IoVs. Dummy location-based privacy preservation methods aim to spam the adversary with fake locations. Accordingly, the user can easily obtain the corresponding answer by filtering out unnecessary results upon receiving all query results from the LBS server. In [46], Dummy-Q, where the query privacy is protected by generating dummy queries, was proposed to preserve the query privacy in the continuous LBS. The privacy leakage problem in the continuous LBS was studied. A frequency-aware dummy-based method (FADBM) was proposed to ensure that dummy locations are generated around frequent areas and the time accessibility [6]. Authors in [7] proposed a dummy filtering algorithm, where the spatiotemporal correlation of time-sensitive side information is used to generate dummy locations. A privacy preservation scheme based on radius constrained dummy trajectory (RcDT) was proposed [8]. By constraining the generated circular range for the location where a user sends an LBS query, the RcDT-based privacy preservation algorithm was proposed to generate the dummy trajectory set with high similarity to the real trajectory comprehensively. Additionally, if an attacker is aware of a specific user’s lifestyle, the attacker can distinguish the user’s location from dummy locations. Several studies attempt to develop dummy location-based techniques for defending against background knowledge attacks by capturing the geographic features of users’ movement or considering both geographic and semantic features [9,10]. Authors in [13] proposed an algorithm to protect location privacy in the continuous LBS, where dummy locations are selected based on the query and transition probabilities. However, as the path length increases, there is an increasing probability that the location information can be inferred by the adversary.

Moreover, obfuscation is adopted to construct the anonymous candidate set for the continuous LBS. The obfuscation is one of the principles used to achieve location privacy by degrading the accuracy of the disclosed location. Most of the obfuscation mechanisms depend on the concept of differential privacy, where privacy is protected by injecting controlled random noise into sensitive data. Geo-indistinguishability presents a practical mechanism for applying differential privacy in LBS [18]. Based on geo-indistinguishability, Authors in [19] proposed an adaptive location preserving privacy mechanism which adjusts the amount of noise required to obfuscate the user’s location. The privacy budget is consumed rapidly, which results in a finite number of LBS queries. Hence, the geo-indistinguishability does not address the potential correlation of the subsequent locations reported within the continuous queries.

A mix zone is considered a promising pseudonym-changing algorithm where mutually cooperative vehicles concurrently change their pseudonym in mix zones. Authors in [30] introduced this concept into the context of LBSs; that is, the mix zone is defined as the spatial region where applications cannot access any location information of users therein. Each user changes the user’s pseudonym to a new and unused one when entering the mix zone. Based on the time when the user enters and leaves the mix zone, an attacker can infer the links between pseudonyms and users. To resist the timing attack, a time window is applied to mix zones [31]. The effectiveness of the mix zone depends on many factors, namely geometry, vehicle density, and geographic location in the road network. Moreover, blocking the use of the LBS inside the mix zone may affect service usage negatively.

Recently, the generative adversarial network (GAN) was applied for trajectory privacy preservation [47]. In [21], the authors envisioned the possibility of leveraging GAN to generate mobility data. Based on [21], a concrete solution was presented in [22] to generate mobility check-in and check-out data, but it cannot generate individual complete mobility trajectories. In [11], a sequence of locations is generated via reinforcement learning and the GAN framework. In [12], two network models were used to decouple spatial information from temporal information in mobility data, which generate location image and timing information separately. However, based on the characteristics of historical query data, the attacker makes trajectory privacy preservation methods invalid using LSA and regional statistical attack (RSA).

Generally, deploying the cache can reduce the information interaction between users and the LBS server, as well as decrease the risk of location privacy leakage. Hence, the cache hit rate is also one of the main performance metrics in the caching-based location privacy protection method. In [23], a collaborative system named MobiCache was proposed to protect users’ location privacy and improve the cache hit rate. A dummy selection algorithm (DSA) was proposed to select dummy locations that have not been cached before to increase the cache hit rate. An entropy-based privacy metric incorporating the effect of caching on privacy was defined in [24]. According to the defined metric, two caching-aware DSAs were developed to enhance location privacy by maximizing the privacy of the current query and the dummies’ contribution to the cache. RuleCache, a multi-level location privacy protection method proposed in [25], combines users’ mobility patterns and utilizes the cache of distributed neighbors to protect location privacy. Moreover, a cloaking region generating algorithm (CRGA) was proposed to protect users’ location privacy, where both query probability and data timeliness are considered. In [26], an enhanced user privacy preservation scheme based on caching and spatial anonymity was proposed, where the multi-level caching mechanism is adopted to reduce the risk of privacy exposure. In [27], a framework enhancing the privacy of LBS using an active caching mechanism was proposed.

Moreover, three broadcasting content selection algorithms, two adaptive updating methods, and one knowledge-based precaching method were addressed. In [28], a multi-hop caching-aware cloaking algorithm was proposed to collect valuable information from multi-hop peers using a collaborative caching mechanism. And a collaborative privacy-preserving querying algorithm was addressed, which sends a fake query to confuse the LBS server. A strategy combining cache scheme with K-anonymous was proposed in [48]. In [29], a blockchain-based privacy-aware content caching architecture was proposed, where the blockchain technology is adopted to record the completed content transactions to solve the problem of distrust between vehicles. However, in IoVs, if the cache is deployed at the user and its neighbors, the validity of the cached data is difficult to guarantee due to the high mobility of vehicles. In this paper, we consider the caching-based location privacy protection method for the continuous LBS, where the cache is deployed at the network edge nodes (i.e., RSUs) in IoVs.

Therefore, according to the system architecture of IoVs, dummy locations and the cache mechanism are used to solve the problem of trajectory privacy protection for the continuous LBS in IoVs in this paper.

## 3. Preliminaries and Problem Formulation

In this section, some preliminaries, including the system model, the LBS query, the service semantics, and the adversary model, are introduced. And the problem aimed to be solved in this work is formulated.

### 3.1. System Model

The architecture of an IoV system, which includes a lot of intelligent vehicles, several RSUs, a trusted authority (TA), and an LBS server, is illustrated in Figure 1. The vehicle-equipped onboard unit (OBU) can acquire the perceived driving information of sensors, calculate, process, and store the sensed data. Moreover, the dedicated short-range communication (DSRC) technology is adopted in the IoV system, which has two communication modes, namely vehicle-to-vehicle (V2V) and vehicle-to-RSU (V2R).

The system architecture shown in Figure 1 is an edge computing network consisting of cloud, edge, and client layers.

Specifically, at the cloud layer, a wide range of infotainment and road-safety-related applications, such as digital map construction and LBS, are supported. RSUs, also “edge nodes” at the edge layer, are deployed at the roadside to speed up the data aggregation and distribution. The edge layer considers how to share data between vehicles and application servers. At the client layer, vehicles acquire data and communicate with their corresponding RSUs and other vehicles.

### 3.2. LBS Query

Let (*x*, *y*) denote the user’s location information, where *x* and *y* represent longitude and latitude, respectively.

For a moving vehicle, its trajectory *Tr* is a set of discrete locations, *Tr* = {*uid*, (*x*_1_, *y*_1_, *t*_1_), (*x*_2_, *y*_2_, *t*_2_), …, (*x_n_*, *y_n_*, *t_n_*)}, where *uid* is a user’s identity, (*x_i_*, *y_i_*, *t_i_*) is the user’s location at time *t_i_*, and *t*_1_ < *t*_2_ < …< *t**_n_*.

An LBS query *Lq* is denoted as *Lq* = {*uid*, (*x_i_*, *y_i_*, *t_i_*), *C*, *V*}, where *C* denotes the user’s querying content sent at location (*x_i_*, *y_i_*) at *t_i_*, *V* is the user’s privacy preservation level, and *V*∈ [0,1). The larger the value of *V*, the more important the privacy is.

### 3.3. Service Semantics

In each location, users may request transportation, entertainment, medical treatment, and/or other services. Service requests sent by users are closely related to their locations, and the probabilities of various services at different locations are different. To represent the relationship between location and service, the service semantics is defined.

Let *U* be the number of services, *e_i,u_* be the request probability of service *u* at location (*x_i_*, *y_i_*), *i* = 0, 1, …, *k* – 1, *u* = 1, 2, …, *U*, and $\sum_{u=1}^{U}e_{i,u}=1$. That is, the service semantics is represented by {*e_i,u_*|*i* = 0, 1, …, *k* – 1, *u* = 1, 2, …, *U*}.

In this work, the LBS server is responsible for the collection and establishment of service semantics.

### 3.4. Adversary Model

The adversary’s goal is to obtain sensitive information about a vehicle user. There are two types of adversary models, namely passive adversary and active adversary.

A passive adversary can monitor and eavesdrop on wireless channels or compromise users to obtain other users’ sensitive information. An eavesdropping attack is performed by the passive adversary to learn extra information about a user.

An active adversary can compromise the LBS server and obtain all the information the server knows. In this work, it is assumed that the LBS server acts as an active adversary. Hence, the adversary can obtain global information and monitor all the LBS queries from users.

In addition, the adversary knows the location privacy preservation scheme adopted in the system. Based on the known information, the adversary tries to infer and learn other sensitive information. Meanwhile, RSUs in IoVs are regarded to be semi-trusted, which means that they normally perform caching and forwarding functions.

According to the trajectory definition, the LBS server can analyze the service request information to get the vehicle user’s locations at different times and arrange the trajectory of the vehicle user according to the time stamps. An RSU may get the vehicle user’s locations within its management area. Since the period of tracking a vehicle user is limited, an RSU cannot obtain the complete trajectory of the vehicle user.

### 3.5. Problem Statement

As the attacker, the LBS server records and classifies historical service requests according to *uid*, using the time stamp and locations in the LBS query to infer the user’s trajectory.

Let *uid*_0_ be the identity of the vehicle user monitored by the LBS server. The trajectory of user *uid*_0_ can be inferred by sorting all service requests {*Lq*} of user *uid*_0_ in chronological order, and *Tr*_0_ = {*uid*_0_, (*x*_1_, *y*_1_, *t*_1_), (*x*_2_, *y*_2_, *t*_2_), …, (*x_n_*, *y_n_*, *t_n_*)}.

For the continuous LBS, the LBS server can perform the LSA and/or LCA to obtain the trajectories of vehicle users.

#### 3.5.1. Long-Term Statistical Attack (LSA)

It is assumed that the map region is divided into several cells.

To protect vehicle location privacy, a location privacy preservation method based on dummy locations, an enhanced-dummy location selection (E-DLS) algorithm [45], is adopted. Once a vehicle user requests an LBS, *k* − 1 dummy locations will be generated. Hence, *Lq* is transformed to *Lq’*, *Lq’* = {*uid*, (*x*, *y*), (*x*_1_, *y*_1_), (*x*_2_, *y*_2_), …, (*x_k_*_−1_, *y_k_*_−1_), *C,*
*C*_1_, *C*_2_,…,*C_k_*_-1_, *V*}, where (*x*_1_, *y*_1_), (*x*_2_, *y*_2_), …, (*x_k_*_−1_, *y_k_*_−1_) are *k* − 1 dummy locations, and *C_i_* represents the user’s querying content sent to a dummy location (*x_i_*, *y_i_*), *i* = 1, 2, …, *k* – 1.

For simplicity, we assume that a vehicle user launches multiple LBS requests in cell O, where *m* cells have the same probability as cell O and can be selected as dummy locations. As the vehicle user sends an LBS query in cell O, the probability of becoming a dummy location in *m* cells is
(1)$$p_{m}=\frac{C_{m-1}^{k-2}}{C_{m}^{k-1}}=\frac{k-1}{m}.$$

At interval *T_ij_* ($T_{ij}=t_{i}-t_{j}$ denotes the time interval between *t_i_* and *t_j_*), it is assumed that the vehicle user launches *f* LBS requests in cell O. In these service requests, *kf* locations are included, where *f* locations are the actual locations of the vehicle user. Let *n*_u_ denote the number of users’ actual locations, and *n*_u_ = *f*. The remaining *k*(*f* − 1) locations are dummy locations.

For *m* cells with the same request probability as cell O, the number of dummy locations to be selected is
(2)$$n_{d}=fp_{m}=\frac{f\left(k-1\right)}{m}.$$

It is assumed that as the vehicle user launches an LBS request in cell O, there are enough cells to be selected as dummy locations. That is, m>k−1. Hence, we have *p_m_* < 1 and *n*_d_ < *n*_u_.

If the user’s actual locations are relatively concentrated, the dummy locations are relatively decentralized because of the randomness and can be ignored by the attacker. Hence, the user’s actual locations cannot be protected by these dummy locations efficiently. In the location privacy preservation method based on dummy locations with the E-DLS algorithm, the LBS server can obtain privacy contents by analyzing historical data. This kind of attack is called LSA [49].

Figure 2 (left) is the vehicle user’s historical service query locations, Figure 2 (right) is the vehicle user’s historical service query locations with the E-DLS algorithm, and *k* = 10. From Figure 2, one finds that although using the location privacy preservation method based on dummy locations, the vehicle location is exposed, and the trajectory cannot be effectively protected under the LSA. This is because the user’s actual locations appear more frequently than dummy locations.

#### 3.5.2. Location Correlation Attack (LCA)

To protect vehicle location privacy, a location privacy preservation method based on spatial K-anonymity is adopted. Once a vehicle user requests an LBS, *Lq* is transformed to *Lq*″, *Lq*″ = (*uid*, {(*x*_0_, *y*_0_), *d*, *C*, *V*}), where (*x*_0_, *y*_0_) is the center of the anonymous region containing *k* users, and *d* is the diameter of the anonymous region. Obviously, the larger the value of *d*, the more uncertain the attacker about the user’s location is, and the better the privacy protection effect is.

As shown in Figure 3, if the attacker connects the centers of multiple anonymous regions, the user’s trajectory is likely to be exposed. This kind of attack is called LCA [50].

Let the anonymous region, including the LBS query sent by user *uid*_0_ at interval *t_i_* be a circle with diameter *d_i_*, and center (*x_i_*, *y_i_*), *i* = 1, 2, …, *n*. Hence, under the LCA, the LBS server can obtain the predicted trajectory of user *u*_0_, *Tr*_0_ = {*uid*_0_, (*x*_1_, *y*_1_, *d*_1_, *t*_1_), (*x*_2_, *y*_2_, *d*_2_, *t*_2_), …, (*x_n_*, *y_n_*, *d_n_*, *t_n_*)}.

If the LBS server acts as an active attacker, it can perform a location-related attack, such as LSA and/or LCA, to obtain the user’s trajectory. Although the location privacy preservation method based on dummy locations or location privacy preservation method based on K-anonymity is adopted, the trajectory privacy cannot be protected effectively. Furthermore, the attacker can infer the user’s other privacy information through the trajectory. Therefore, the problem of trajectory privacy protection should be solved urgently.

## 4. Proposed Trajectory Privacy Preservation Method Based on Caching and Dummy Locations

In this section, a vehicle trajectory privacy preservation method based on caching and dummy locations, abbreviated as TPPCD, is presented. TPPCD consists of a dummy locations generation algorithm, a cache initialization mechanism, a cache updating scheme, and an LBS query-response mechanism. Since the RSU can obtain the location of the vehicle user within its coverage by physical signal-based means, we assume that the vehicle user’s location is transparent to the RSU. Moreover, it is assumed that an RSU is semi-trusted.

### 4.1. Parameter Setting

Before sending the LBS query, the vehicle user should set the corresponding privacy protection level *V* according to the privacy protection requirement.

In the proposed method, due to the caching mechanism, the RSU will not forward the received LBS query to the LBS server if the cached data are hit. When the cached data are missed, the transformed LBS query is sent to the LBS server using dummy locations. Hence, the success rate of vehicle location privacy protection is:(3)$$V\geq \gamma +\left(1-\gamma \right)\left(1-\frac{1}{k}\right),$$
where *γ* denotes the threshold of the cache hit rate. The right side of the inequality in (3) represents the success rate of vehicle location privacy protection if the caching locations at the RSU are randomly selected. However, to improve the cache hit rate, the hotspot data are usually selected to be cached at the RSU. Hence, the success rate of vehicle location privacy protection of the proposed method is higher than the value of the right side of the inequality in (3).

Considering the worst-case scenario, to guarantee the vehicle location privacy, the privacy parameter *k* is determined by the vehicle location privacy protection level *V,* and the threshold of the cache hit rate. In the proposed method, privacy parameter *k* denotes one LBS query sent by a vehicle user containing *k* − 1 dummy locations. That is,
(4)$$k=\lceil \frac{1-\gamma }{1-V}\rceil ,$$
where ⌈⋅⌉ denotes the upper integer operation.

### 4.2. Cache Initialization Mechanism

The LBS server divides the region covered by an RSU into *I*×*J* cells. *cell_i_*_,*j*_ denotes the cell of row *i* and column *j*, *i* = 1, 2, …, *I*, and *j* = 1, 2, …, *J*. The location of *cell_i_*_,*j*_ can be denoted as ***r****_i_*_,*j*_, and ***r****_i_*_,*j*_ = (*x_i_*_,*j*_, *y_i_*_,*j*_), which is a location randomly selected within *cell_i_*_,*j*_. Let the service request probability of *cell_i_*_,*j*_ be *q_i,j_*, the service semantics of service *u* at *cell_i_*_,*j*_ be *e*_(*i*,*j*),*u*_. The service request probability *q_i,j_* means the probability that the user initiates a request in *cell_i_*_,*j*_ among all cells of the region covered by an RSU. And *e*_(*i*,*j*),*u*_ means the probability that the user initiates service type *u* among all requested service types in *cell_i_*_,*j*_.

At the initialization phase, based on the collected historical service data, the LBS server constructs the information matrix *Q*(***r***, ***q***, ***e***) consisting of the locations of cells, service request probability ***q***, and service semantics ***e***. And then, the LBS sends the information matrix *Q*(***r***, ***q***, ***e***) to the RSU.

Meanwhile, the LBS server actively pushes part of the hotspot data to the RSU. The hotspot data are the request results of ⌈γIJU⌉ locations and service combinations with the highest request probability in the coverage of the RSU.

The problem of hotspot data locations selection can be formulated as
(5)$$\underset{D}{\max}\sum q_{i,j}e_{(i,j),u}\;s.t.D(r,q,e)\subset Q(r,q,e)\;\forall r_{i,j}\in D,r_{i,j}\in C,r_{i,j}\in R\;u\in \{1,2,3,\cdots ,U\}\;\left|D\right|=\lceil \gamma IJU\rceil ,$$
where **R** represents the location area accessible by the road. D is the set of hotspot locations and service contents, *D*(***r***, ***q***, ***e***) is the corresponding information matrix. C is the set of all locations of cells in the area covered by the RSU.

To solve the problem formulated in (5), the LBS server selects ⌈γIJU⌉ hotspot data in a greedy manner.

According to *Q*(***r***, ***q***, ***e***), the LBS server calculates the probability of service request at each location in **R**, $q_{i,j}e_{(i,j),u}$, *i =* 1, 2, …, *I*, *j =* 1, 2, …, *J*, *u =* 1, 2, …, *U*, $cell_{i,j}\in R$. The LBS server chooses ⌈γIJU⌉ hotspot data with the largest probability of service requests through ⌈γIJU⌉ rounds.

In the *l*th round, *l* = 1, 2, …, ⌈γIJU⌉, the LBS server selects the locations and service to D from set {$q_{i,j}e_{(i,j),u}$} which maximizes the sum of the probability of service requests in D, and deletes it from set {$q_{i,j}e_{(i,j),u}$}.

Hence, set D is constructed with the hotspot locations and service contents.

After receiving the information matrix and the hotspot data, the RSU caches the date and constructs the cached information matrix E(r,q,t,e) based on *Q*(***r***, ***q***, ***e***) and the existing time $t_{(i,j),u}$ of cached data. The information matrix of the locations without cached data is *P*(***r***, ***q***, ***e***). The RSU broadcasts *Q*(***r***, ***q***, ***e***), **R** and γ to vehicle users in its coverage.

### 4.3. Dummy Locations Generation Algorithm

The vehicle user uses the dummy location selection algorithm under road restriction (RR-DLS) [51] to generate dummy locations.

Suppose set G includes *k* locations, and $G=\{(x_{0},y_{0}),(x_{1},y_{1}),\ldots ,(x_{k-1},y_{k-1})\}$. The anonymous entropy is defined as
(6)$$H=-\sum_{i=0}^{k-1}p_{i,u}\log_{2}p_{i,u},$$
where
(7)$$p_{i,u}=\frac{q_{i}e_{i}{}_{,u}}{\sum_{i=0}^{k-1}q_{i}e_{i,u}}.$$

In this algorithm, entropy is used to represent the degree of anonymity. The effective distance represents the minimum distance between the current location and other locations in a location set. The dummy location selection algorithm aims to maximize the anonymous entropy and the effective distance of the candidate location set consisting of the vehicle user’s location and dummy locations, ensuring the uncertainty and dispersion of selected dummy locations.

### 4.4. Cache and Information Matrix Update Mechanism

RSU needs to request the LBS server to realize the cache update. Hence, in this section, we consider using dummy locations to protect the location privacy of vehicle users and using the service results on dummy locations to realize the passive cache update. Since there are different types of LBS services in IoVs, and the query data of different services in different locations have different contributions to the cached data, dummy locations should be selected according to the validity of the cached data and privacy protection in corresponding locations, as well as protecting location privacy and improving the cache hit rate. Meanwhile, since the information matrix is considered in the RR-DLS, it should be updated to ensure its effectiveness.

The probability of different service types at different locations is different. The higher the request probability of a specific service at a specific location, the higher probability of being hit. Moreover, as content to be cached at an RSU keeps longer, the validity of cached data goes weaker. Hence, the existing time, $t_{(i,j),u}$, is used to measure the validity.

The vehicle user generates dummy locations using the RR-DLS to protect the location’s privacy. G is the location and corresponding service set, and |G|=k. Since the service query generated by the vehicle user contains multiple locations and service contents, there will be some queries without cache. Assume that the number of RSU cache hits in the location set is *k*_c_. If the service requested by the vehicle user’s service query is hit at RSU, the corresponding $t_{(i,j),u}$ will be reset to 0.

When $k_{c}/k\geq \gamma$, the cached data will not be updated temporarily, and the RSU will return the service query results at *k*_c_ locations. When the returned service query results contain the result required by the vehicle user, the vehicle user gets the service. Otherwise, when the returned service query results do not contain the result required by the vehicle user, the vehicle user sets the NoCache identifier in the service request $Lq^{\prime }$, transforms the service request $Lq^{\prime \prime \prime }$ as $Lq^{\prime \prime \prime }=\left\{Lq^{\prime },\mathrm{NoCache}\right\}$, and sends $Lq^{\prime \prime \prime }$ to the RSU.

When $k_{c}/k<\gamma$ or NoCache identifier is set in the service query, the cached data and the information will be updated. The RSU generates *k*_c_ dummy locations to replace the locations hit by the cache, constructs a service request *Lq*^*^ and sends it to the LBS server.

The problem of *k*_c_ dummy locations selection at the RSU can be formulated as
(8)$$\underset{ℬ}{\max}\left\{-\sum p_{(i,j),u}\log_{2}p_{(i,j),u}+H_{n}\right\}\;s.t.B(r,q,e)\subset P(r,q,e)\;\forall r_{i,j}\in B,r_{i,j}\in C,r_{i,j}\in R\;\left|ℬ\right|=k_{c},$$
where
(9)$$p_{(i,j),u}=\frac{q_{i,j}e_{(i,j),u}}{\sum_{r_{i,j}\in ℬ,u\in ℬ}q_{i,j}e_{(i,j),u}+\sum_{r_{i,j}\in G^{\prime },u\in G^{\prime }}q_{i,j}e_{(i,j),u}},$$
(10)$$H_{n}=-\sum_{r_{i,j}\in G^{\prime },u\in G^{\prime }}^{}p_{(i,j),u}\log_{2}p_{(i,j),u},$$
where ℬ is the set of *k*_c_ dummy locations and corresponding services generated at the RSU, B(r,q,e) is the information matrix corresponding to set ℬ, and $G^{\prime }$ is the set of locations and corresponding services missed by the cache in G.

To solve the problem formulated in (8), the RSU selects *k*_c_ locations whose service request probabilities are closed to locations in $G^{\prime }$.

After receiving *Lq*^*^, the LBS server return the results to the RSU. Due to the caching deployment at the RSU, the actual location of the vehicle user may not be included in the service query *Lq*^*^, which confuses the LBS server.

To cache data effectively, it is preferential that the cached data are replaced by data not requested for a long time. Hence, according to the cached information matrix E(r,q,t,e), the RSU caches the results of *Lq*^*^ and deletes the cached data whose existing times are larger than *T*_D_, where *T*_D_ denotes the lifetime of cached data.

Finally, the RSU sends the corresponding query results of G to the vehicle user.

To update the information matrix, the RSU sends the number and type of services provided by the RSU using cached data to the LBS server during a period of *T*.

The LBS server recalculates the service query probability and service query semantics of each cell and updates the information matrix. Whenever the information matrix is updated, the LBS server sends the information matrix *Q*(***r***, ***q***, ***e***) to the RSU.

The RSU updates the cached information matrix E(r,q,t,e) based on *Q*(***r***, ***q***, ***e***). As in the initialization step, the RSU broadcasts the updated information matrix *Q*(***r***, ***q***, ***e***) to the vehicle user.

### 4.5. LBS Query Response Mechanism

When a vehicle user requests an LBS, *k* − 1 dummy locations are generated using RR-DLS. When the RSU receives the transformed LBS query sent by the vehicle user, it returns the response of the LBS query if the querying contents are hit.

The RSU should construct a new LBS query based on the querying contents missed by the cache and send it to the LBS server. The LBS server returns the response of the LBS query. The result of the user’s query is obtained by interacting with the LBS server. Hence, the vehicle user receives the results of the LBS query from the RSU.

### 4.6. Overall Procedure of TPPCD

Taking the example of a vehicle user sending requests, the procedure of the TPPCD is composed of the initialization phase and the user’s query phase.

The procedure of TPPCD is summarized as follows.

Initialization Phase:

(1)The LBS server collects the number and types of services provided by the RSU using cached data and counts the number of various types of service requests sent by vehicle users in each cell. Hence, for each location, the LBS server can calculate the corresponding historical query probability $q_{i,j}=\frac{f_{i,j}}{F}$, where $f_{i,j}$ denotes the number of queries in location *cell_ij_*, *F* is the total number of service requests in the area under the jurisdiction of RSU. The request probability of service *u* is $e_{(i,j),u}=\frac{f_{(i,j),u}}{f_{i,j}}$, where $f_{(i,j),u}$ is the number of queries of service *u* in location *cell_ij_*, *u* = 1, 2, …, *U*. The LBS server constructs the information matrix *Q*(***r***, ***q***, ***e***). Moreover, the LBS server selects the hotspot data based on the results of problem (5). Finally, the LBS server sends the information matrix *Q*(***r***, ***q***, ***e***) and the hotspot data to the RSU.(2)The RSU broadcasts information such as *Q*(***r***, ***q***, ***e***), **R** and γ to vehicle users in its jurisdiction and constructs a cached information matrix E(r,q,t,e) based on the existing times of cached data. Meanwhile, the RSU begins to count the existence time *t_q_* of the information matrix.(3)The vehicle users store the information matrix according to the broadcast information.

User’s Query Phase:

(1)Using the privacy protection level *V* and the threshold of cache hit rate γ, the vehicle user calculates the privacy parameter *k* according to Equation (4). As described in Section 4.3, the vehicle user generates *k* – 1 dummy locations with the information matrix *Q*(***r***, ***q***, ***e***) and road information **R**, and RR-DLS. And a service query Lq′ is constructed and sent to the RSU.(2)After receiving Lq′, the RSU retrieves the cached data for service queries. The number of queries hit by the cached data is *k*_c_. The existing times of hit data are reset to 0 s. If $k_{c}/k\geq \gamma$, the service results are returned directly to the vehicle user. Otherwise, if $k_{c}/k<\gamma$, it goes to Step (4).(3)The vehicle user filters the service results based on its location. If the user’s real query result is obtained, it goes to Step (8). Otherwise, the vehicle user sets the NoCache identifier, constructs the service query Lq‴ and sends it to the RSU.(4)The RSU selects *k*_c_ locations and their corresponding services according to the result of the problem (8). The RSU replaces the queries hit by the cached data in Lq′ with the locations in set ℬ obtained in problem (8) and the corresponding services to construct the service query Lq*. The RSU sends Lq* to the LBS server.(5)Receiving Lq*, the LBS server retrieves the database and returns the service results to the RSU.(6)The RSU caches the results of *Lq** and deletes the cached data whose existing times are larger than *T*_D_. Then, the RSU screens out the results required by the vehicle user and adds the queries hit by the cached data in Step (2) to construct the user’s service results and returns them to the vehicle user.(7)The vehicle user filters the service result according to its location.(8)If $t_{q}\geq T$ , the RSU perform the information matrix update step as described in Step 1).(9)The whole service for a vehicle user is finished.

## 5. Performance Evaluation and Discussion

In this section, the performance of TPPCD is analyzed in terms of security, the computation overhead, the communication overhead, and the storage overhead. Moreover, the performance of TPPCD is evaluated. In addition, we compare the performance of the proposed method with some existing methods based on cache or dummy locations.

The data set used in simulations was collected from 182 users in the Microsoft Research Asia Geolife project. The GPS trajectory is represented by a sequence of points with time stamps, each containing latitude, longitude, and altitude information. The data set contains 17,621 trajectories, with a distance of 12,92951 km and a total duration of 50,176 h. It is widely distributed in over 30 cities in China, even in some cities in the United States and Europe, but most data were created in Beijing, China [52].

### 5.1. Security

Since encrypt-based technologies can be easily applied to the proposed TPPCD method, eavesdropping attacks on wireless channels between vehicle users and other entities can be ignored. We focus on LSA and LCA from the LBS server as an active adversary.

#### 5.1.1. Long-Term Statistical Attack

When a vehicle user sends a service query to the RSU, the location set contains *k* − 1 dummy locations. Hence, the probability that the RSU identifies the actual location of a vehicle user from the location set is 1/*k*.

If the RSU needs to update the cache, it selects dummy locations according to the anonymous entropy and replaces the locations hit by the cached data. In this process, the vehicle user does not interact with the LBS server, and the location set generated by the RSU for the cache update does not necessarily contain the vehicle user’s actual location, which confuses the LBS server. The probability that the LBS server identifies the actual location of a vehicle user from the location set is still 1/*k* or even 0. The returned query results are cached in the RSU, which further reduces the probability of the user’s location leaked to the LBS server.

Suppose a vehicle user uses the TPPCD method to initiate *f* LBS queries at cell O in a period. The number of cells with the same service query probability of cell O is *m*. The probability of cell O being selected is *p*_c_, and the probability of a cell with the same request probability being selected as a dummy location is *p*_m_. On the LBS server side, the number of times the vehicle user’s actual location contained is *n*_u_, and the number of dummy locations is *n*_d_. We have
(11)$$n_{u}\leq (1-\gamma )f+p_{c}\gamma f,$$
(12)$$n_{d}\leq (1-\gamma )f\frac{k-1}{m}+p_{m}\gamma f.$$

From (11) and (12), one finds that the ratio between *n*_u_ and *n*_d_ cannot be determined as *m* and *γ* are large. That is, with the deployment of cache in the RSU, the probability of the actual location of a vehicle user in the historical data of the LBS server is not necessarily larger than that of the dummy locations.

From the above proofs, the trajectory of a vehicle user cannot be obtained effectively through LSA using the TPPCD method.

To verify the privacy protection performance of our proposed algorithm, we do a few simulations.

Figure 4 shows the trajectory data of vehicle users used in simulations, the east longitude ranges from 116.295° to 116.345°, and the north latitude ranges from 39.935° to 39.995°. The area is 6000 m × 3500 m.

The above area is divided into 200 × 100 cells, and the service request probabilities corresponding to the cells under different privacy preservation methods are counted and calculated. The total number of service requests in the area is 2780.

Figure 5a shows the service request probability of vehicle users without a location privacy preservation mechanism. It can be seen from Figure 5a that the service request probability distribution is consistent with the trajectory of vehicle users.

Figure 5b shows the probability distribution of service requests with the location privacy preservation method based on dummy locations. From Figure 5b, the location privacy preservation method based on dummy locations is used. The service request probability distribution is fuzzy, but the probability of service requests on dummy locations is much smaller than on the actual location. Hence, the trajectory information of vehicle users can still be distinguished according to the probability of service requests.

Figure 5c shows the probability distribution of service requests with TPPCD. From Figure 5c, owing to the caching mechanism, the vehicle user reduces the number of service requests sent to the LBS server directly. The number of service requests sent to the LBS server is 569. Moreover, due to the dummy query introduced by the caching mechanism and cache update, the service query is further blurred. The service request probability distribution obtained by the LBS server cannot obtain the trajectory information related to the vehicle user.

#### 5.1.2. Location Correlation Attack

As shown in Figure 6, the theoretical premise of LCA is that there is a spatial-temporal correlation between locations, and each anonymous area or location set should contain the actual location of a vehicle user.

In TPPCD, since the dummy locations set sent to the LBS server may not contain the vehicle user’s actual location, the spatial-temporal correlation of location information received by the LBS server is guaranteed. Hence, it is difficult to obtain the trajectory of a vehicle user through LCA. Next, we will prove these analyses through simulation compared with the K-anonymity method [34] and dummy location-based method [45].

Figure 7 shows a part of the vehicle user trajectory in Figure 4. The east longitude ranges from 116.298° to 116.312°, the north latitude ranges from 39.983° to 39.993°, the area is 1100 m × 1200 m, four RSUs are set in the area, and each RSU covers a range of 500 m × 500 m. The time interval between two adjacent locations in the figure is 30 s. The vehicle’s true trajectory is shown as the blue line, and the blue number denotes the location sequence in the trajectory.

Figure 8 shows the predicted trajectory of the vehicle user when the location privacy preservation method based on K-anonymity is used, where *k* = 10. In Figure 8, the green circle represents the anonymous area, and the green number denotes the location sequence in the true trajectory corresponding to the anonymous area. Figure 9 shows the predicted trajectory of the vehicle user when the location privacy preservation method based on dummy locations is used, where *k* = 10. In Figure 9, the purple dot represents the dummy location and the purple number dennotes the location sequence in the true trajectory corresponding to the dummy locations set.

From Figure 8 and Figure 9, with the location privacy preservation method based on K-anonymity or dummy locations, each location of the vehicle user is replaced by an anonymous area. However, the vehicle trajectory predicted by the LCA is close to the vehicle’s true trajectory in Figure 7, which means that the vehicle trajectory information is not protected effectively.

Figure 10 shows the predicted trajectory of the vehicle user when TPPCD is used, where *V* = 0.95. The cache hit rate threshold *γ* is 0.5 and 0.75 in Figure 10a,b, respectively. In Figure 10, the gray dot represents the dummy location in the query sent by the RSU, and the gray number denotes the location sequence in the true trajectory corresponding to the dummy locations set generated by TPPCD. From Figure 10, the cache introduced by TPPCD reduces the information interaction between the vehicle user and the LBS server, and the dummy query is introduced into the cache update mechanism. Hence, the trajectory of the vehicle user predicted by LCA differs from the true trajectory, as shown in Figure 7. The results show that the LBS server cannot obtain the vehicle user’s trajectory through LCA, and the user’s trajectory information is protected effectively.

To further illustrate the effectiveness of vehicle trajectory privacy preservation methods, the predicted location deviations for three location privacy preservation methods are calculated. With the K-anonymity-based location privacy protection method, the estimated location deviation is about 41.55 m. With a dummy locations-based location privacy protection method, the estimated location deviation is about 49.38 m. With TPPCD, the estimated location deviation is about 134.92 m and 286.76 m for *γ* = 0.5 and *γ* = 0.75, respectively. Hence, the proposed vehicle trajectory privacy preservation method, TPPCD, can resist LCA effectively.

### 5.2. Cache Hit Rate

In this simulation, four location privacy preservation methods, E-DLS in [45], CaDSA in [24], RuleCache in [25], and PAPT in [29], are compared with TPPCD proposed in this paper. The cache hit rate refers to the probability that the demand can be satisfied by cached data at the RSU when the vehicle user requests the service and is defined as the ratio of the number of services obtained by the user through the RSU to the total number of requested services.

Figure 11 shows the cache hit rate of different location privacy preservation methods based on the cache. The cache data lifetime *T*_D_ = 3 h, the information matrix update period *T* is also 3 h, and the interval of the service requests is 10 min. From Figure 11, since E-DLS does not adopt cache, the cache hit rate is 0. PAPT is based on an active cache mechanism, the LBS server pushes the service contents to the RSU for caching, and the cache hit rate is about 0.9. Three other methods are based on passive cache mechanisms, where the cache hit rate increases gradually over time. CaDSA deploys the cache at the vehicle user, and the cache hit ratio is about 0.7. RuleCache establishes caches in the user’s local area, neighbor users, and the LBS server, and the cache hit rate is about 0.9 after stabilization. TPPCD deploys the cache on the RSU, and the cache hit rate is about 0.9. In the proposed method, combined with the active cache mechanism, the LBS server sends the hotspot data to the RSU in the cache initialization stage to reach the cache hit rate threshold. Moreover, combined with the cache update mechanism based on dummy locations, the cache hit rate keeps high. Compared with PAPT, TPPCD simplifies the active caching strategy, reduces the complexity of the caching mechanism but still guarantees a high cache hit rate.

### 5.3. Service Failure Rate

In the simulation, we use the frequency of the NoCache identifier is used to characterize the service failure rate.

Figure 12 shows the service failure rate of TPPCD, where *V* = 0.95, the total number of service requests is 2780, and the frequency of the NoCache identifier is calculated every 200 times. From Figure 12, we observe that when *γ* = 0.5, the frequency of the NoCache identifier fluctuates between 4% and 12%, with an average value of 8.54%. When *γ* = 0.75, the frequency of the NoCache identifier fluctuates between 2% and 10%, with an average value of 5.87%. The reason for this phenomenon is that as the cache hit rate threshold increases, the number of hotspot data caches increases, the cache hit rate increases, and the corresponding service failure rate decreases. When the vehicle user cannot obtain the desired result from the cached data at the RSU, the NoCache identifier will be set to the service request, which results in increasing the communication overhead of the vehicle user and the service delay.

### 5.4. Overhead Analysis

#### 5.4.1. Computation Overhead

If the coverage of an RSU is divided into *I* × *J* cells, the number of services is *U* and the number of POIs returned by the LBS server is *n*.

In the procedure of an LBS query, since the vehicle user uses RR-DLS to select dummy locations, the computation overhead at the vehicle user is 𝒪(*k*^2^ + *IJU*).

The computation overhead of the RSU is divided into two parts, dummy locations selection and caching queries. Dummy locations are selected based on the validity of the cache; the computation overhead is 𝒪(*IJU*). The query operation is to retrieve the cached data. The worst case is to traverse all the cached data, and the computation overhead is 𝒪(*IJU*). Therefore, the computation overhead at the RSU is 𝒪(*IJU*).

The LBS server needs to perform service retrieval for *k*−1 dummy locations and a real location to obtain *kn* service results. Hence, the computation overhead at the LBS server is 𝒪(*kn*).

#### 5.4.2. Communication Overhead

In the procedure of an LBS query, a vehicle user only sends its actual location and *k*−1 dummy locations with corresponding query contents to the RSU. The communication overhead at the vehicle user is 𝒪(*k*).

For the RSU, there are two situations.

When $k_{c}/k\geq \gamma$, the RSU directly sends the service results hit by cached data to the vehicle user. When the service request cache is hit, the communication overhead is the largest and 𝒪(*kn*).

When $k_{c}/k<\gamma$ or the RSU receives the LBS query with the NoCache identifier, the RSU generates dummy locations according to the cache update mechanism and then sends a service request to the LBS server. The communication overhead is 𝒪(*k*). After the LBS server returns the query results, the RSU integrates the results and returns them to the vehicle user. The communication overhead is 𝒪(*kn*).

Therefore, the communication overhead of the RSU is 𝒪(*k + kn*).

The communication overhead of the LBS server is divided into two parts.

In the cache initialization stage, the hotspot data are sent to the RSU. The communication overhead is 𝒪(⌈γIJU⌉n).

Receiving the service query, the LBS server returns the corresponding service results to the RSU. The communication cost is 𝒪(*kn*).

Therefore, the communication overhead at the LBS server is 𝒪(⌈γIJU⌉n+kn).

#### 5.4.3. Storage Overhead

Due to the cache deployment at the RSU, the additional storage overhead is 𝒪(*IJUn*).

## 6. Conclusions

In this paper, we studied the vehicle trajectory privacy-preserving problem for the continuous LBS in IoVs. A trajectory privacy preservation method based on caching and dummy locations, TPPCD, is proposed. In the proposed method, the vehicle user-generated dummy locations protect the vehicle location when it acquires the continuous LBS. Moreover, the cache is deployed at the RSU to reduce the information interaction between vehicle users covered by the corresponding RSU and the LBS server. In the proposed cache update mechanisms, data popularity and dummy locations are considered to protect location privacy and improve the cache hit rate. The RSU caches the hotspot data pushed by the LBS server in the initialized and periodic cache update phase and the requested data sent by the LBS server in the service-providing phase. The performance analysis and simulation results show that the proposed vehicle trajectory privacy preservation method can resist LSA and LCA and protect the vehicle trajectory privacy effectively, as well as guaranteeing a high cache hit rate. However, the proposed method increases the computation overhead and communication overhead at the RSUs and the LBS server.

This study considered the trajectory privacy protection for the continuous LBS, but did not consider data utility for subsequent trajectory data research. Moreover, it is assumed that an RSU is semi-trusted. Hence, in future research, we will concentrate on the following aspects: (1) Considering the trade-off of trajectory privacy protection and data utility. (2) Further consider the protection of RSU’s possible privacy leakage. (3) Continuous LBSs entail more user queries and higher real-time requirements, resulting in a greater challenge involving the cache hit ratio than that of snapshot LBS, which leads to more communication overhead. Reducing communication overhead is also a research direction.

## Figures and Tables

**Figure 1 sensors-22-04423-f001:**
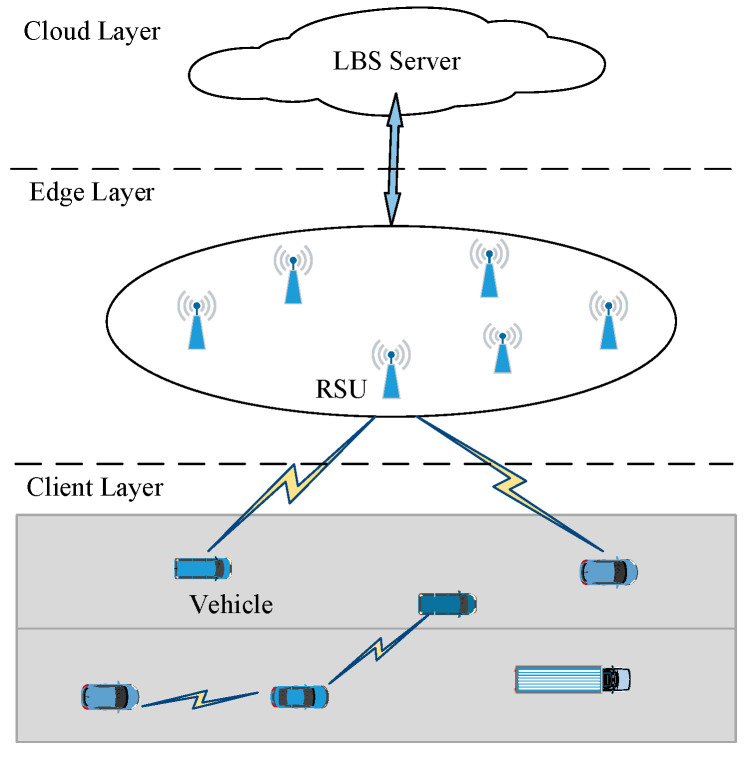
The architecture of an IoV system.

**Figure 2 sensors-22-04423-f002:**
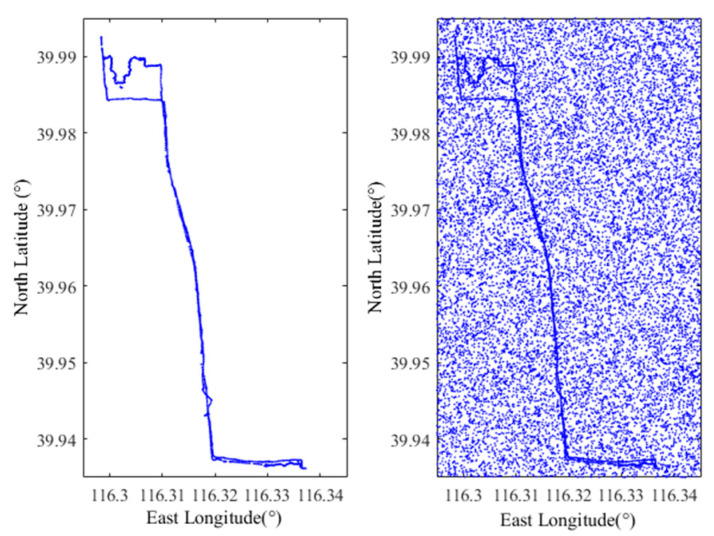
The schematic of long-term statistical attack (LSA).

**Figure 3 sensors-22-04423-f003:**
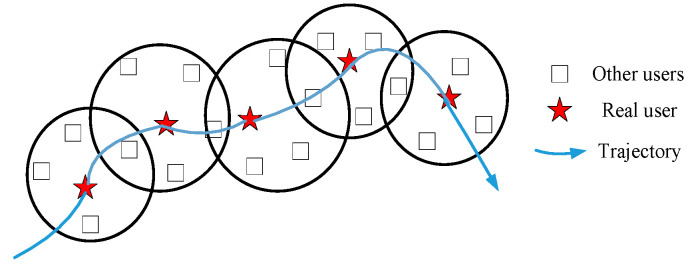
The schematic of location correlation attack (LCA).

**Figure 4 sensors-22-04423-f004:**
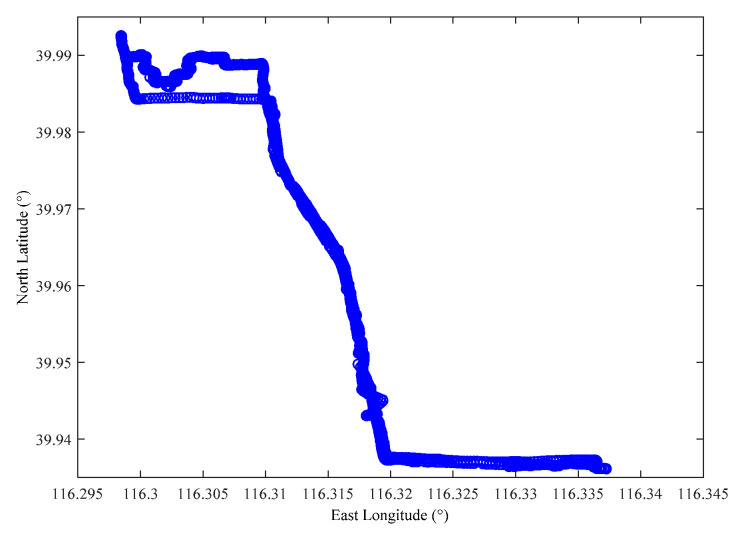
The schematic diagram of vehicle user trajectory.

**Figure 5 sensors-22-04423-f005:**
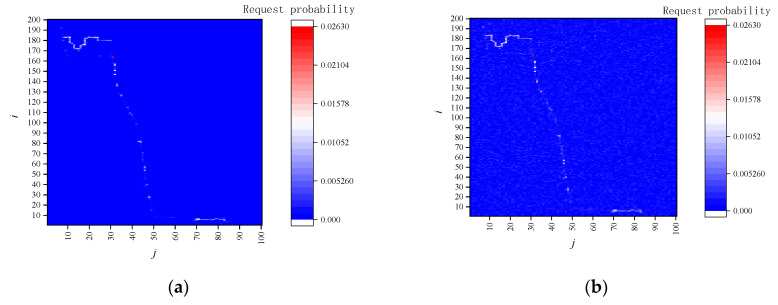
The probability distribution of service requests. (**a**) without location privacy preservation method; (**b**) with location privacy preservation method based on dummy locations; (**c**) with the proposed trajectory privacy preservation method.

**Figure 6 sensors-22-04423-f006:**
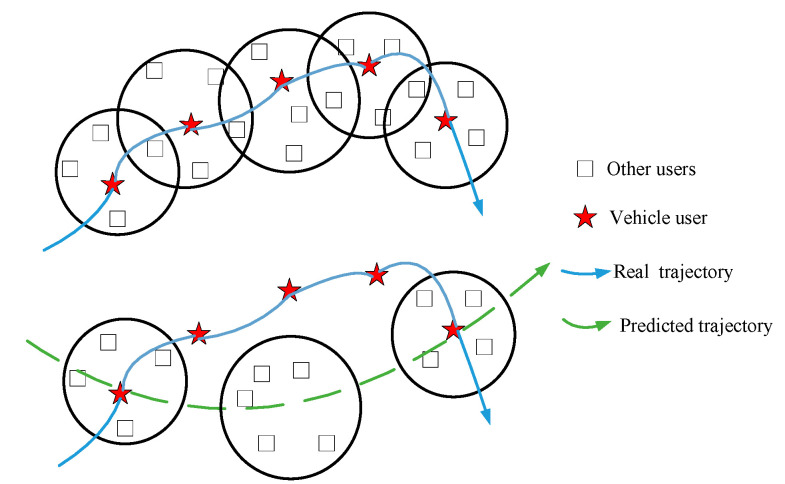
Resisting location correlation attack (LCA).

**Figure 7 sensors-22-04423-f007:**
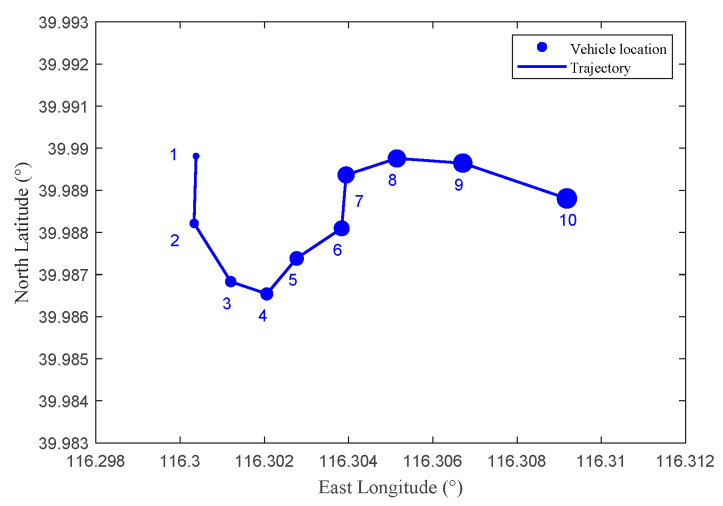
The true trajectory of a vehicle user without location privacy preservation method.

**Figure 8 sensors-22-04423-f008:**
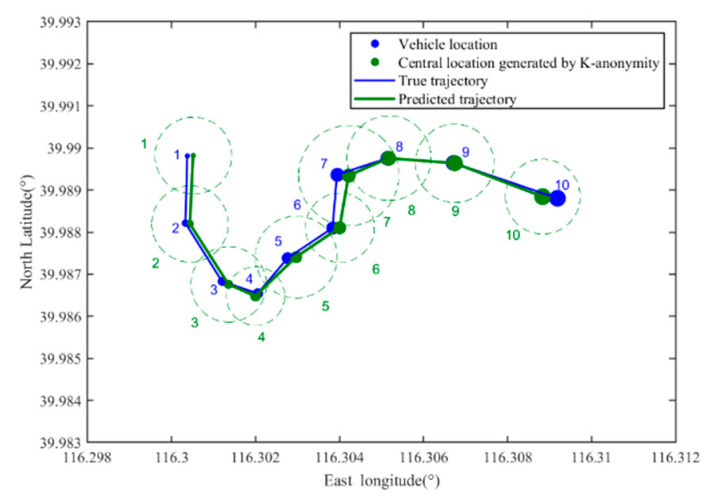
The predicted trajectory of a vehicle user with location privacy preservation method based on K-anonymity.

**Figure 9 sensors-22-04423-f009:**
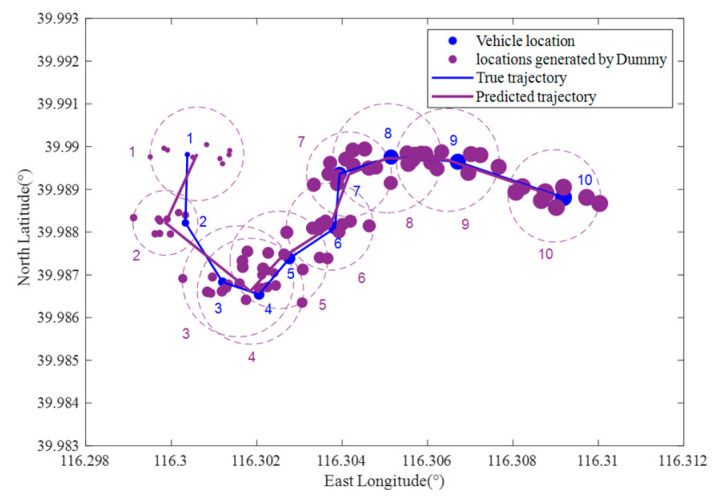
The predicted trajectory of a vehicle user with location privacy preservation method based on dummy locations.

**Figure 10 sensors-22-04423-f010:**
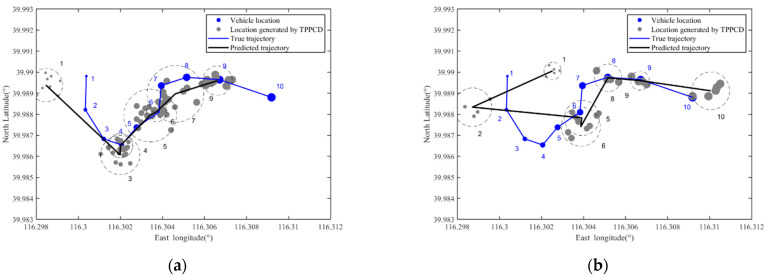
The predicted trajectory of a vehicle user with TPPCD. (**a**) V=0.95 γ=0.5; (**b**) V=0.95 γ=0.75.

**Figure 11 sensors-22-04423-f011:**
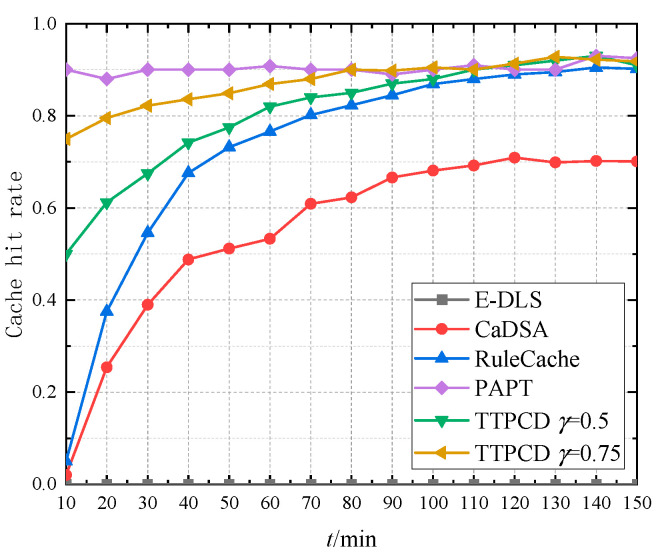
The cache hit rate.

**Figure 12 sensors-22-04423-f012:**
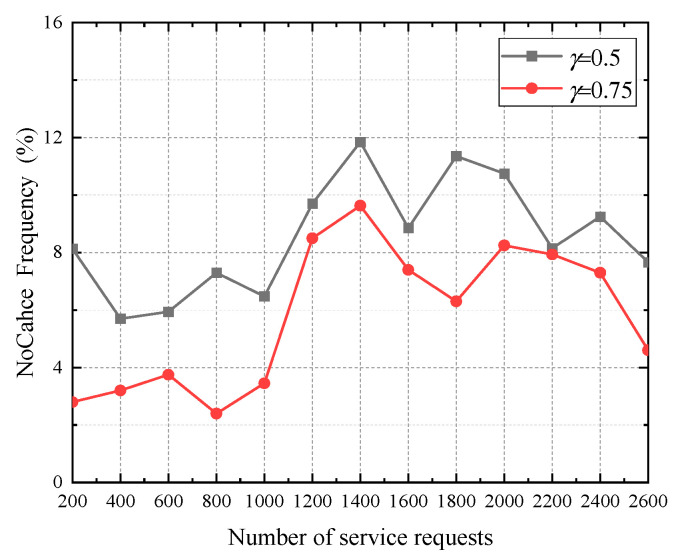
The service failure rate of TPPCD.

## Data Availability

Not applicable.

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
