# Peer review of "A Vehicle Trajectory Privacy Preservation Method Based on Caching and Dummy Locations in the Internet of Vehicles"

_sensors, 2022, doi:10.3390/s22124423_

Round 1

Reviewer 1 Report

The subject of the article is interesting and worth describing. However, the method of implementation requires correction. In the Introduction, the authors presented an introduction to the subject. The Introduction section is deficient. It does not contain all the necessary elements. The main goal of the research and specific goals were not clearly defined. The Introduction section should contain research hypotheses or research questions.
The layout of the work is not entirely correct. I have already listed what should be in section 1 Introduction. The Materials and Methods section is missing. It should be included in the article as section 3. This section should contain information about the research period and more detailed information about the scope of the research. This information is included in sections 3 and 4. These sections can be subsections within the Materials and Methods section. In the Materials and Methods section, there is space to specify the test steps.
There is no clearly defined Results section. This section may contain the results from the current section 5 and some from section 6.
There should be a separate Discussion section. It was formed as the 6. Performance Evaluation and Discussion. This section needs to be revised. Some of the material may be in the Results section. The discussion itself should be placed in a separate Discussion section. I understand a discussion as referring to other studies after presenting my research results. In my opinion, doing research without a clear comparison and reference to other research results in the fact that the obtained results cannot be properly assessed. Such references are. However, there is no broad scientific discussion.
Conclusion section is incomplete. The conclusions can be bulleted. You must certainly refer to the hypotheses set at work. Have they been verified positively or negatively? Future research directions should also be provided.
I analyzed the bibliography. The scientific discussion is quite poor. Articles by authors from China were mainly referenced. There is no broad scientific discussion of an international scope.

Other remarks:
The abbreviation LBS is used in the abstract. Its meaning was explained only in the next sentence. This should have been done when using the shortcut for the first time.
The title of section 5 is Performance Analysis. Analysis is a research method. The title should be more substantive.

Reviewer 2 Report

In this paper, the authors proposed a vehicle trajectory privacy preservation method based on caching and dummy locations. The work is solid. Some suggestions are presented as follows. 

1. The authors mentioned "The main contributions of this paper are as follows". The authors are suggested to highlight the main differences between the proposed work and the existing published works.

2. The physical meanings of all symbols and notation should be explained more in detail.

3. Section 4.7 should be explained more in detail about each step.

Reviewer 3 Report

This paper proposes a vehicle trajectory privacy preservation method based on caching and dummy locations, abbreviated as TPPCD, in IoVs. In the proposed method, when a vehicle user wants to acquire a continuous LBS, the dummy locations-based location privacy preservation method under road constraint is used. The idea looks good. Here are some comments:

1.     What is the full name of TPPCD? What is the definition of “Dummy Locations” in this paper?

2.     How to get the vehicle trajectory if these data sets are treated as private information?

3.     It is not necessary to have “Overview” subsection.

4.     The related work is not sufficient. Many latest references, such as Smart Collaborative Tracking for Ubiquitous Power IoT in Edge-Cloud Interplay Domain, Location Privacy Preservation Based on Continuous Queries for Location-Based Services, should be introduced and compared.

5.     “The communication overhead is O(k).” how to prove it?

Round 2

Reviewer 3 Report

The current version is fine.